# High-throughput micro-CT analysis identifies sex-dependent biomarkers of erosive arthritis in TNF-Tg mice and differential response to anti-TNF therapy

H. Mark Kenney [1,2], Kiana L. Chen[1,2], Lindsay Schnur[1], Jeffrey I. Fox[1], Ronald W. Wood[1,3,4,5], Lianping Xing[1,2], Christopher T. Ritchlin[1,6], Homaira Rahimi[1,7], Edward M. Schwarz[1,2,5,6,8,9], Hani A. Awad[1,8,9]*

1 Center for Musculoskeletal Research, University of Rochester Medical Center, Rochester, NY, United States of America, 2 Department of Pathology & Laboratory Medicine, University of Rochester Medical Center, Rochester, NY, United States of America, 3 Department of Obstetrics and Gynecology, University of Rochester Medical Center, Rochester, NY, United States of America, 4 Department of Neuroscience, University of Rochester Medical Center, Rochester, NY, United States of America, 5 Department of Urology, University of Rochester Medical Center, Rochester, NY, United States of America, 6 Department of Medicine, Division of Allergy, Immunology, Rheumatology, University of Rochester Medical Center, Rochester, NY, United States of America, 7 Department of Pediatrics, Pediatric Rheumatology, University of Rochester Medical Center, Rochester, NY, United States of America, 8 Department of Biomedical Engineering, University of Rochester Medical Center, Rochester, NY, United States of America, 9 Department of Orthopaedics, University of Rochester Medical Center, Rochester, NY, United States of America

* Hani_Awad@urmc.rochester.edu

**Data Availability Statement:** Data is publicly available via the Zenodo repository: Kenney, H. M., Chen, K. L., Schnur, L., Fox, J. I., Wood, R. W.,

## Abstract

### Background

Development of reliable disease activity biomarkers is critical for diagnostics, prognostics, and novel drug development. Although computed tomography (CT) is the gold-standard for quantification of bone erosions, there are no consensus approaches or rationales for utilization of specific outcome measures of erosive arthritis in complex joints. In the case of pre-clinical models, such as sexually dimorphic tumor necrosis factor transgenic (TNF-Tg) mice, disease severity is routinely quantified in the ankle through manual segmentation of the talus or small regions of adjacent bones primarily due to the ease in measurement. Herein, we sought to determine the particular hindpaw bones that represent reliable biomarkers of sex-dependent disease progression to guide future investigation and analysis.

### Methods

Hindpaw micro-CT was performed on wild-type (n = 4 male, n = 4 female) and TNF-Tg (n = 4 male, n = 7 female) mice at monthly intervals from 2–5 (females) and 2-8-months (males) of age, since female TNF-Tg mice exhibit early mortality from cardiopulmonary disease at approximately 5-6-months. Further, 8-month-old WT (n = 4) and TNF-Tg males treated with anti-TNF monoclonal antibodies (n = 5) or IgG placebo isotype controls (n = 6) for 6-weeks were imaged with micro-CT every 3-weeks. For image analysis, we utilized our recently

Xing, L., Ritchlin, C. T., Rahimi, H., Schwarz, E. M., & Awad, H. A. (2024). Data for: High-throughput micro-CT analysis identifies sex-dependent biomarkers of erosive arthritis in TNF-Tg mice and differential response to anti-TNF therapy [Data set]. Zenodo. https://doi.org/10.5281/zenodo.11191782. For any questions related to the data, contact the University of Rochester's Miner Library Data Services (miner_information@urmc.rochester.edu).

**Funding:** This work was supported by funding from the National Institutes of Health (NIH): F30AG076326 (HMK), T32GM007356 (HMK), T32AR076950 (HMK, KLC), R01AR069000 (CTR), R01AR056702 (EMS), and P30AR069655. HMK is a trainee in the Medical Scientist Training Program funded by NIH T32GM007356. The content is solely the responsibility of the authors and does not necessarily represent the official views of the National Institution of General Medical Science or NIH. The funders did not play any role in the study design, data collection and analysis, decision to publish, or preparation of the manuscript.

**Competing interests:** The authors have declared that no competing interests exist.

developed high-throughput and semi-automated segmentation strategy in Amira software. Synovial and osteoclast histology of ankle joints was quantified using Visiopharm.

## Results

First, we demonstrated that the accuracy of automated segmentation, determined through analysis of ~9000 individual bones by a single user, was comparable in wild-type and TNF-Tg hindpaws before correction (79.2±8.9% vs 80.1±5.1%, $p = 0.52$). Compared to other bone compartments, the tarsal region demonstrated a sudden, specific, and significant bone volume reduction in female TNF-Tg mice, but not in males, by 5-months (4-months 4.3 ± 0.22 vs 5-months 3.4± 0.62 mm$^3$, $p<0.05$). Specifically, the cuboid showed significantly reduced bone volumes at early timepoints compared to other tarsals (i.e., 4-months: Cuboid -24.1±7.2% vs Talus -9.0±5.9% of 2-month baseline). Additional bones localized to the antero-lateral region of the ankle also exhibited dramatic erosions in the tarsal region of females, coinciding with increased synovitis and osteoclasts. In TNF-Tg male mice with severe arthritis, the talus and calcaneus exhibited the most sensitive response to anti-TNF therapy measured by effect size of bone volume change over treatment period.

## Conclusions

We demonstrated that sexually dimorphic changes in arthritic hindpaws of TNF-Tg mice are bone-specific, where the cuboid serves as a reliable early biomarker of erosive arthritis in female mice. Adoption of automated segmentation approaches in pre-clinical or clinical models has potential to translate quantitative biomarkers to monitor bone erosions in disease and evaluate therapeutic efficacy.

## Introduction

Establishment of sensitive and specific biomarkers of disease activity is essential for reliable diagnostics, prognostics, and novel drug development. However, the determination of interpretable and quantitative outcomes is a complicated process in conditions with complex and multifactorial pathologic mechanisms. For instance, rheumatoid arthritis (RA) exhibits autoimmunity and systemic inflammation that localizes to particular joints with synovitis and bone erosions that promote severe articular deformities, dysfunction, and pain. At its core, the hallmark of RA pathogenesis is immune dysregulation associated with a multitude of downstream processes, including inappropriate responses to self-antigens (predominately post-translationally modified epitopes) [1] and defective stromal cells in the setting of chronic inflammation (i.e., pathologic synovial fibroblasts [2–5] or reduced joint-draining lymphatic function [6–8]). In fact, the diversity of pathologic processes that compose RA have been proposed to represent a "syndrome" with enough clinical overlap to represent a distinct clinical entity, but inevitably exhibits patient-specific presentations dependent on factors such as genetic predisposition and epigenetic influences [1, 9]. Given the various mechanisms that mediate RA pathogenesis, a crucial question remains–how do we accurately and reliably monitor clinical progression or response to therapy for an individual patient?

The current standard of care practices for diagnosis and follow-up of RA rely predominately on a combination of serum markers and clinical evaluation. Initial laboratory tests

typically consist of autoantibody assessment (i.e., rheumatoid factor and anti-citrullinated protein antibodies) and general indicators of inflammation (i.e., erythrocyte sedimentation rate [ESR] and C-reactive protein [CRP]). While the discovery of RA-associated autoantibodies has provided highly specific biomarkers for diagnostic purposes, these factors exhibit relatively low sensitivity accounting for the prevalence of seronegative RA [10]. On the other hand, measurements of ESR and CRP exhibit sensitivity for identification of inflammation without specificity to RA. Many additional serum biomarkers of RA have been proposed [11], including a multibiomarker disease activity score [12–15]. Such resources may provide valuable advances in future disease monitoring, but as of the most recent American College of Rheumatology (ACR) recommendations for RA disease activity measures, the benefits of these options for clinical use remain inconclusive [16].

Instead, ACR recommendations are primarily based on clinical assessment via disease activity scores (DAS) in RA. Such measures typically include a quantification of total tender and swollen joints along with patient self-interpretation of current disease status, which are often combined with ESR or CRP inflammatory markers to quantify progression of arthritic severity (i.e., DAS28-ESR/CRP) [16]. While these approaches provide clinical insights and value to characterize global effects of RA on patient health in an efficient and cost-effective manner, these scoring strategies are limited due to the subjective characterization by patients and potential for variability in reliable interpretations of tender or swollen joints between different providers [17]. In addition, quantification of total affected joints may not accurately characterize pathology at specific articular surfaces, and inaccurately perceive the true severity of a particular patient's disease course. Thus, limitations remain possible given the semi-quantitative nature of clinical disease activity scores where joint-specific or subclinical arthritis may be mischaracterized or undetectable, respectively.

While clinical disease activity indices are beneficial, there remains an inconsistency in how these scores correlate with the actual pathophysiological manifestations, such as localized inflammation in joints or erosion in particular bones. To augment clinical interpretation, various imaging techniques have been utilized, most notably ultrasound (US), magnetic resonance imaging (MRI), and conventional X-ray, to provide additional quantitative evaluation of disease severity even at subclinical levels. For instance, recent evaluation of joint US revealed that patients considered to be in remission based on DAS-28 scoring systems repeatedly identified subclinical synovitis [18, 19], and US was found to be a better predictor of future radiologic progression than clinical assessment [20]. While US provides the benefit of rapid imaging and quantifiable outcomes (i.e., synovial volume, power doppler blood flow), there are mixed reports regarding the correlation of US outcomes with clinical measures [21, 22], likely related to a deficiency in considering the complete patient presentation and inter-operator reliability (i.e., image quality, transducer pressure) [23]. MRI also provides the potential for quantitative metrics with exquisite detail to detect changes in soft tissue inflammation (i.e., synovitis, tenosynovitis) and bone damage (i.e., erosions, edema). In fact, reliable scoring systems, such as RA MRI score (RAMRIS), have been developed to determine severity of such specific features of RA disease [24]. However, the remarkable imaging time of MRI at a resolution necessary for longitudinal volume tracking of distinct structures limits the feasibility of routine disease activity monitoring. Lastly, conventional X-ray imaging has been a mainstay of monitoring RA progression through well-described scoring systems for identification of peri-articular erosions and joint-space narrowing representing cartilage destruction [25].

Beyond these imaging techniques, innovative approaches towards the utilization of computed tomography (CT) in RA diagnostics and monitoring disease activity have been limited. CT exhibits remarkable capacity to spatially resolve and identify bone pathology given the distinct density of bone [25], and novel dual energy CT methods have even been developed to

concurrently assess surrounding soft tissue [26]. Although an important drawback and consideration of CT is exposure to ionizing radiation, these effects are relatively limited in distal extremities frequently involved in RA (i.e., hands, feet) [25] and image reconstruction techniques for improved utilization of low-dose CT imaging methods are also being developed [27]. CT is considered the standard reference for evaluation of bone towards both identification and quantification of erosion (primarily via manual segmentation) [28–31], but to our knowledge there have been no CT-specific consensus methods to monitor bone erosions during RA progression [16, 25]. However, efforts towards this goal are underway [32, 33]. Such advances in imaging technologies to monitor early RA radiographic changes are valuable as more recent considerations of RA pathogenesis and therapeutics highlight the potential for a subclinical "pre-RA" phase where timely intervention could prevent disease [1, 34].

Pre-clinical models also provide tremendous promise for the development and optimization of validated and high-throughput image analysis techniques with opportunities for adoption into clinical medicine. Our prior work has focused on assessment of arthritis progression using the well-characterized tumor-necrosis factor transgenic (TNF-Tg) murine model of RA [35, 36]. We have utilized US [37, 38], MRI [37, 39], and micro-CT [39, 40] to evaluate the progression of the spontaneous inflammatory-erosive arthritis and associated pathologic processes, which exhibit many similarities with human RA, including certain extra-articular manifestations [41–43] and a remarkable sexual dimorphism with earlier progression of disease in female mice [44]. In our experience, quantification of bone volumes and erosions using micro-CT exhibited the greatest optimization of efficiency and inter-user reliability in analysis of arthritic progression compared to other techniques, despite the inability to simultaneously measure associated soft-tissue inflammation *in vivo*. However, the analysis of bone erosions in complex joints, such as the hindpaw (30–31 bones in C57BL/6 mice [45, 46]), exhibited remarkable complications, leading to this simple question–which bone should be analyzed? In both pre-clinical and clinical investigation, this question is of immense importance as bone-specific susceptibility to pathologic processes or treatments inevitably affects the interpretation of disease progression where "incorrect" bone selection could unknowingly lead to underestimation of disease severity.

The development of high-throughput and reliable strategies to quantify progression of bone-specific erosive arthritis using CT imaging has the potential to provide remarkable advances for disease activity scoring methods. Previously, in pre-clinical arthritis models, disease severity has been routinely quantified in the ankle through manual segmentation of the talus or small regions of adjacent bones [39, 47]. Recently, we developed a novel semi-automated and efficient segmentation method to quantify each individual bone volume of healthy wild-type (WT) murine hindpaws from micro-CT datasets with excellent inter-user reliability [46]. In the current work, we demonstrate the capacity to translate this approach to evaluate bone volumes in TNF-Tg hindpaws with progressive and severe erosive arthritis towards rigorous identification of bone-specific, temporal, sex-dependent, and treatment-responsive (anti-TNF) biomarkers of disease progression to guide future investigation.

## Materials and methods

### Ethical approval

All animal experiments were performed on IACUC protocols approved by the University Committee on Animal Resources at the University of Rochester Medical Center and mice were housed within an AAALAC accredited vivarium.

## Mouse models

Wild-type (WT) and TNF-Tg mice (3647 line, C57BL/6 background) [36, 48, 49] were originally obtained from Dr. George Kollias [36, 48, 49] and have been since maintained at the University of Rochester. TNF-Tg mice were bred as heterozygotes and were genotyped using the following primer sets:

TNF-Tg Forward: 5`-TAC-CCC-CTC-CTT-CAG-ACA-CC-3`
TNF-Tg Reverse: 5`-GCC-CTT-CAT-AAT-ATC-CCC-CA-3`

A total of 32 mice were used for this study. Starting at 2-months of age, 19 mice (n = 4 WT male, n = 4 WT female, n = 4 TNF-Tg male, and n = 7 TNF-Tg female) had ankle micro-CT images collected at monthly intervals. Following the 5-month images, the female cohorts were euthanized to collect joints for histology as female TNF-Tg mice are known to exhibit early mortality compared to males between approximately 5-6-months of age [44]. In fact, 2 TNF-Tg females unexpectedly died during the course of the study, one before 4-months and another before 5-months of age. Micro-CT images at monthly intervals continued to be collected in male mice until 8-months of age. At 8-months of age, at which time male mice begin to exhibit onset of severe inflammatory-erosive arthritis [50, 51], 7 additional TNF-Tg male mice were included from our previous study [50] (permission for reuse described below in Data Availability). The TNF-Tg mice were matched into placebo (IgG1 isotype control antibodies; CNTO151 Janssen, J&J; n = 6) and anti-TNF (CNTO12 Janssen, J&J; n = 5) treatment groups (intraperitoneal, 10 mg/kg/week) based on their lymph node volume [50], a biomarker of arthritic progression [38, 39, 51–53], while WT mice (n = 4) received volume-matched vehicle PBS. The mice were also concurrently administered bromodeoxyuridine (intraperitoneal, 0.1 mg/kg/day) for endpoints associated with a separate study [50]. The mice were treated for a total of 6-weeks, as the time period previously shown to ameliorate TNF-mediated disease [43, 50, 54], with ankle micro-CT images collected at 3-week intervals during the treatment period. Given the well-established asymmetry of TNF-Tg arthritis [38, 39, 44, 50, 52–55], individual limbs were used as the unit of measure in this study.

## Micro-CT data collection

Ankle micro-CT images were collected as previously described [46, 50, 56]. Briefly, mice were anesthetized with 1–3% isoflurane and imaged using a VivaCT 40 (Scanco Medical, Bassersdorf, Switzerland) for 30–45 minutes in a Derlin plastic and clear acrylic tube. Datasets were imaged with the following parameters: 55 kV, 145 µA, 300 ms integration time, 2048 x 2048 pixels, 1000 projections over 180˚, resolution 17.5 µm isotropic voxels. The DICOM files were then exported for downstream analysis.

## Micro-CT image segmentation and analysis

For the hindpaw image segmentation, we utilized our recently described high-throughput semi-automated segmentation protocol within Amira software (v2020.2; ThermoFisher Scientific, FEI, Hillsboro, OR, USA) [46]. In brief, this segmentation method utilizes the original image input (following application of a median filter), a set threshold binary mask of bone (>2500 Hounsfield units), and the semi-automated generation of material seeds that serve as markers of the individual bones. Using these inputs, a "Marker Based Watershed Inside Mask" module is applied, which expands the bone-specific markers automatically to the edges of each specific bone. The end result is a completely segmented hindpaw where volume measurements can be extracted for each of the 30–31 bones present in a mouse hindpaw. The variability in number of bones (30–31) is due to the sporadic fusion of the intermediate cuneiform with the navicular / lateral cuneiform (navicular and lateral cuneiform are consistently fused in C57BL/

6 mice [45, 46]) that even occurs within animals, where in one limb the bones may be fused and the other they are separate. Given the variable fusion, these bones were combined together for a single volume measure in this study, and are referred to in shorthand as NAVLATINT (fused navicular, lateral cuneiform, and intermediate cuneiform). Additional shorthand codes for the bones analyzed in this study are listed here: Metatarsal (Met), proximal phalange (PP), distal phalange (DP), sesamoid (S), calcaneus (Calc), cuboid (Cub), medial cuneiform (Med), talus (Tal), and tibiale (Tib). For the bones associated with the digits (Met, PP, DP, and S), the bones are numbered in increasing order from medial to lateral. In total, this segmentation method has allowed for the quantitative analysis of approximately 9,000 individual bones by a single user.

The WT mice analyzed in this study were used to describe the segmentation method [46] (permission for reuse described in Data Availability), while this is the first report of the novel methodology applied to TNF-Tg hindpaws with severe erosive arthritis. We calculated the error rate by counting the number of bones in each dataset that were segmented incorrectly as a percentage of total bones, and described methods were utilized to fix any errors [46]. Errors were categorized as oversplit (1 bone split into >1 material), overconnected (>1 bone combined into 1 material), or a combination of both error types. Following segmentation of each bone, a "Material Statistics" module was used to extract volumes for each of the individual materials representing distinct bones. Calculations of percent change utilize 2-months of age as the baseline, when the first micro-CT scans were collected, unless otherwise noted. In total, 3 datasets were excluded from the study due to motion artifact that reduced the capacity for accurate segmentation and volume measurements (1 WT male at 2-months (and thus time-points dependent on 2-month baseline measurement), 1 WT female at 3-months, 1 TNF-Tg female at 4-months). A demonstration of the analysis output in WT and TNF-Tg mice, as well as a description of the distinct error types, is provided in Fig 1.

## Histology

Following euthanasia with a lethal ketamine/xylazine cocktail and secondary cervical dislocation, the ankles were isolated and surrounding soft tissue removed. The ankles were then fixed in 10% neutral buffered formalin for 3 days and decalcified in Webb-Jee 14% EDTA solution for 1 week before being processed for paraffin embedding. For each bone analyzed, sections at 5 μm from 3 levels were collected. To compare bone specific biomarkers, the tibiale, cuboid, and talus were targeted specifically for sectioning (S1 Fig). Sections were then stained for H&E-OG and TRAP to evaluate synovium and osteoclasts, respectively. Joints from an additional 3 WT and 3 TNF-Tg male mice at 5-6-months of age were similarly processed to compare with female histologic findings during this time period.

All slides were imaged with a VS120 Slide Scanner and imported into Visiopharm software (v2021.07; Horsholm, Denmark) to perform semi-automated color segmentation of the different stains (S2 Fig), as previously described [44, 50, 51, 57, 58]. The tissue area was defined by the H&E-OG sections, which included predominately bone with small contributions of adjacent soft tissues. For the brightfield images, analysis was performed on a section at each of the 3 levels then outcomes averaged. There were n = 3 samples in which only 2 levels were averaged as the bone of interest could not be identified. There was n = 1 hindpaw (TNF male) that was excluded from analysis because the talus was not present in the sections, while there was n = 1 hindpaw (TNF female) excluded from the cuboid analysis for both H&E-OG and TRAP staining due to identification as an outlier. For the female tibiale-targeted histology, n = 3 WT and n = 3 TNF-Tg hindpaws were excluded as the tibiale was lost while optimizing the sectioning approach.

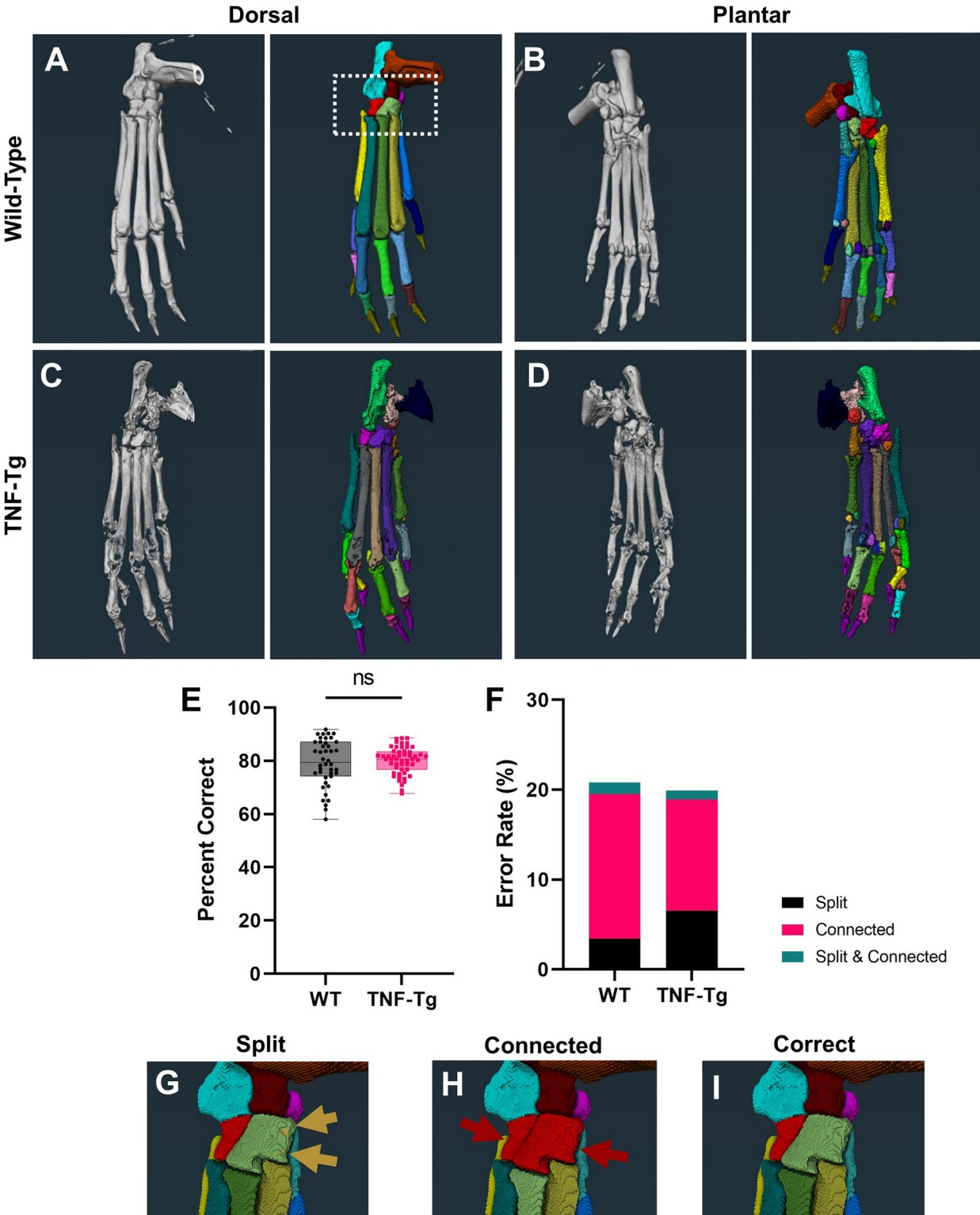

**Fig 1. Semi-automated segmentation of TNF-Tg hindpaws with severe arthritis shows comparable accuracy to wild-type counterparts.** To evaluate bone-specific erosions in the TNF-Tg mouse model of inflammatory-erosive arthritis, we utilized our recently published high-throughput semi-automated hindpaw segmentation protocol [46]. Representative images of the dorsal and plantar surfaces of hindpaw micro-CT images (left) and the segmentation of each bone indicated by unique colors (right) are provided for wild-type **(A-B)** and TNF-Tg **(C-D)** male mice at 8-months of age. Without user intervention, the semi-automated protocol produces accurate segmentations of approximately 80% of bones in wild-type

datasets (error rate ~20%), which remarkably remains consistent, and potentially improved through reduced variance, for TNF-Tg mice with bone erosions (**E**). Note that TNF-Tg segmentations have increased split and reduced connected errors, corresponding with eroded bones (**F**). If present, these errors are corrected utilizing described workflows [46] to produce the final correct segmentation. Examples of errors and correction (**G-I**) are provided from a high-magnification image of the tarsal region in the wild-type hindpaw shown in **A** (white box).

## Statistics

All statistical analysis, such as unpaired t-test and 2-way ANOVA or mixed-effects analysis with Tukey's multiple comparisons, were performed in GraphPad Prism (v9.5.0; San Diego, CA, USA). Outliers were identified using the ROUT method with Q = 1%. To evaluate biomarker potential in the treatment groups, measures of effect size were calculated between the TNF-Tg placebo and anti-TNF treated mice using outcomes of the 2-way ANOVA analysis across time. The following equations were used to calculate eta-squared ($\eta^2$)

$$\eta^2 = \frac{SSeffect}{SStotal}$$

partial $\eta^2$

$$partial\ \eta^2 = \frac{SSeffect}{SStotal + SSerror}$$

and omega-squared ($\omega^2$)

$$\omega^2 = \frac{SSeffect - DFeffect*MSerror}{SStotal + MSerror}$$

where SS = sum of squares, DF = degrees of freedom, MS = mean of squares. Effect indicates the condition analyzed (treatment x time), total is the sum for all conditions, and error refers to the residual.

## Results

### TNF-Tg females exhibit accelerated bone loss with temporal transition from phalange to tarsal erosions

To evaluate for potential biomarkers of erosive arthritis in the TNF-Tg hindpaws, we first assessed whether our previously described high-throughput semi-automated segmentation method [46] would accurately segment bones in samples with disease and deformity. As such, resultant representative images of the original micro-CT datasets and complete bone segmentation are provided from the dorsal and plantar perspective for both WT and TNF-Tg male mice at 8-months of age (Fig 1A–1D). Remarkably, the TNF-Tg hindpaws were segmented at a similar accuracy to WT samples (WT 20.8%, TNF-Tg 19.9% error), and intuitively the phenotypic erosions aided in reducing connected errors (WT 16.1% vs TNF-Tg 12.4% error), although at the expense of increasing split errors (WT 3.4% vs TNF-Tg 6.5%) (Fig 1E and 1F). Of note, the TNF-Tg samples also exhibited significantly reduced variance in segmentation accuracy compared to WT (Fig 1E, *p = 0.0002*). Examples of split and connected errors were artificially generated to demonstrate the presentation of these segmentation mistakes compared to the correct identification of each individual bone in a high-magnification of the tarsal region (box in Fig 1A and 1G–1I).

Through segmentation of each bone between WT and TNF-Tg hindpaws, we organized the bone types into distinct compartments (Tarsals, red; Mets, blue; PP, yellow; DP, white; and S, green [plantar surface]) to assess erosive patterning over time during progression of inflammatory-erosive arthritis (Fig 2A–2D). While total bone volume significantly decreased over time

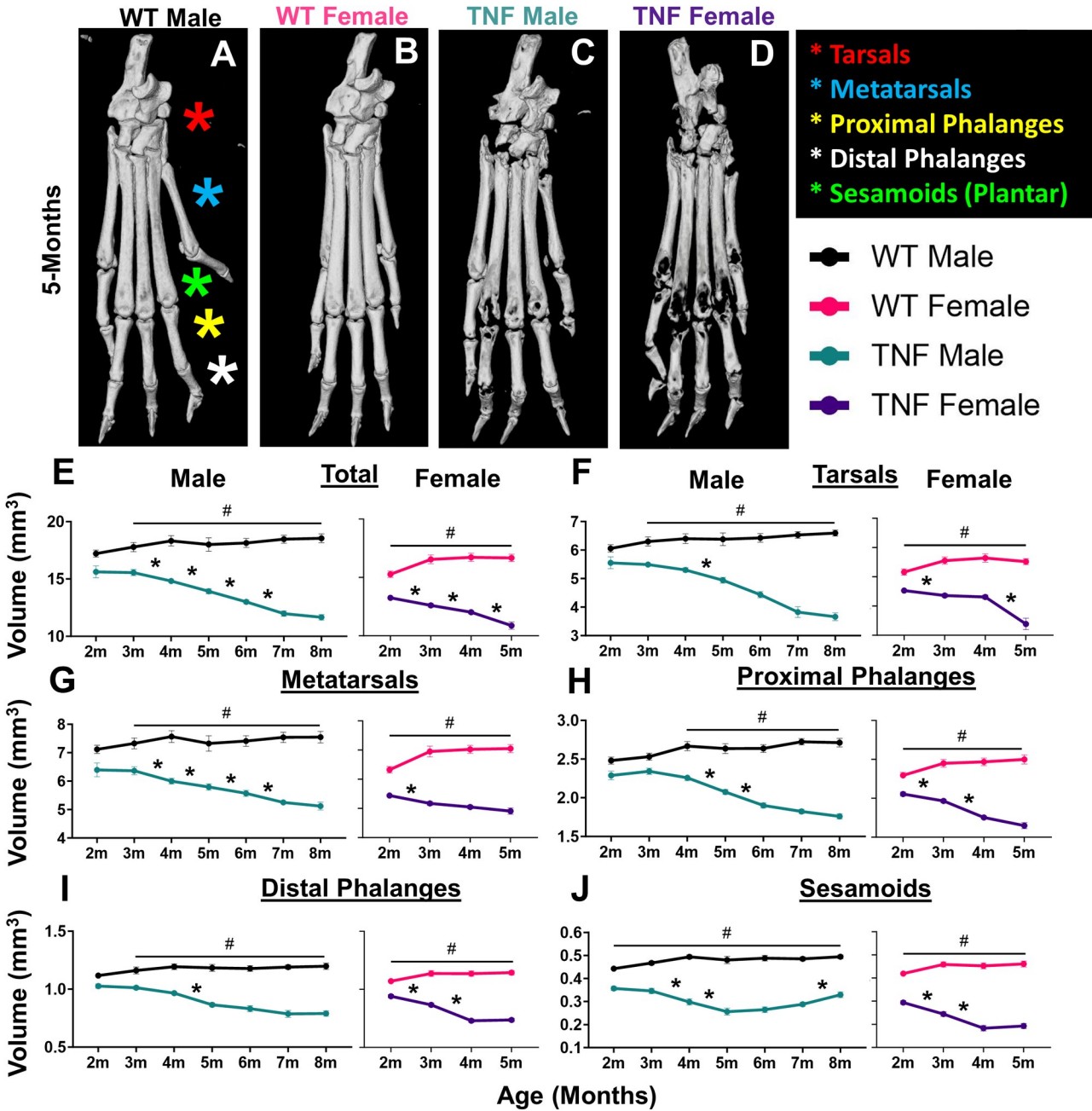

**Fig 2. TNF-Tg females exhibit accelerated bone loss with temporal transition from phalange to tarsal erosions.** Patterns of bone loss in particular hindpaw compartments (tarsals, red; metatarsals, blue; proximal phalanges, yellow; distal phalanges, white; and sesamoids, green) were analyzed for temporal and genotype effects. Representative images of hindpaws at 5-months of age are shown for wild-type males (starting sample size, n = 6 hindpaws), wild-type females (n = 8 hindpaws), TNF-Tg males (n = 8 hindpaws), and TNF-Tg females (n = 14 hindpaws) with the compartments labeled by colored stars **(A-D)**. Although both TNF-Tg males and females exhibited consistent bone loss across time **(E, *p<0.05, between timepoints)**, TNF-Tg females showed unique time-dependent erosions by particular compartment. For TNF-Tg females, there was a dramatic decrease in the bone volume of the tarsals between 4-5-months of age **(F)** that was preceded by early erosions starting at 2-months of age in the bones associated with the phalanges that was not sustained past 4-months **(G-J)**. On the other hand, males showed relatively slow progression of erosions with limited statistical change for particular compartments at monthly intervals, except for the metatarsals **(F-J)**. Also note the "U"-shaped progression of bone volumes in the TNF-Tg male sesamoids, likely representing bone remodeling and fusion following an initial period of erosions **(J)**. Compared to wild-type mice, both TNF-Tg male and female mice showed significantly reduced bone volumes in all compartments starting at 3-months and 2-months of age, respectively (#p<0.05). Statistics: 2-way ANOVA (males) and mixed-effects analysis (females) with Tukey's multiple comparisons **(E-J)**.

in both male and female TNF-Tg mice compared to WT (Fig 2E), we noted compartment specific changes. The TNF-Tg male mice tended to exhibit gradual erosions across bone compartments along with a notable "U"-shaped trajectory of the sesamoid bones where erosions were followed by remodeling and fusion. On the other hand, female TNF-Tg mice exhibit early erosions specific to the bones associated with the digits (Mets, PPs, DPs, S). This initial decline in bone volume is followed by a period from 4-5-months of age where the digit volumes remain stable (i.e., DPs: 4-months $0.73\pm0.05$ vs 5-months $0.74\pm0.04$ mm$^3$, *p>0.05*), while the tarsal bones demonstrate a sudden, specific, and significant reduction in bone volume during this time same time period (Fig 2F–2J; 4-months $4.3\pm 0.22$ vs 5-months $3.4\pm 0.62$ mm$^3$, *p<0.05*). The unique compartment changes in bone volume across time for each group are summarized as heatmaps, which further highlights bone-specific severe erosions (Fig 3, white/ yellow = increased bone volumes, purple/black = decreased bone volumes). Representative images of the total hindpaw and high-magnification images of each individual compartment for all groups across time are provided in S3–S9 Figs. Note the repetition of 5-month images from Fig 2A–2D and S3 Fig to visualize these hindpaws in the context of changes over time.

## The cuboid is an early biomarker of erosive arthritis in TNF-Tg female mice

Given the unique decline in bone volume within the tarsal compartment for TNF-Tg females, we performed a focused evaluation of bone-specific changes that may be driving this pattern of disease and serve as a biomarker of arthritic severity. High-magnification images of the tarsal region in 5-month-old mice for all groups are provided, and each specific bone is highlighted (Calc, black; Cub, pink; Med, green; NAVLATINT, dark purple; Tal, light purple; Tib, blue) (Fig 4A–4D). Note the repetition of 5-month images from Fig 4A–4D in S1 and S5 Figs for visualization of the tarsals in the context of histologic sectioning and changes over time, respectively. To directly compare volume changes across time, percent change in bone volume was calculated through normalization to the 2-month baseline. TNF-Tg male mice exhibited no difference in bone volume change between the individual bones, until at 8-months old where the tibiale demonstrated higher volumes compared to all other bones except the medial cuneiform (*p<0.05*, Fig 4E). Overall, the cuboid showed the earliest decline in bone volume, while by 6-8-months-old both the talus and cuboid demonstrated comparable, consistent, and most severe erosive activity in male mice, although notably not significantly different compared to other tarsals. TNF-Tg female mice instead revealed unique bone-specific changes across time, where at 3- and 4-months of age the cuboid showed significantly reduced bone volumes compared to all other bones, except medial cuneiform at 3-months (4-months: Cuboid $-24.1\pm7.2\%$ vs Talus $-9.0\pm5.9\%$ change in bone volume, comparing the top 2 bones with the greatest change, *p<0.05*). However, between 4-5-months of age, both the talus (4-months $-9.0\pm5.9\%$ vs 5-months $-40.8\pm20.3\%$) and NAVLATINT (4-months $-8.6\pm4.0\%$ vs 5-months $-31.9\pm15.0\%$) demonstrated a remarkable decline in bone volume, reaching similar values as the cuboid (4-months $-24.1\pm7.2\%$ vs 5-months $-44.9\pm15.0\%$), which are responsible for the sudden decline in tarsal bone volumes during this time period noted in Fig 2F (Fig 4F). To further investigate the cellular mechanisms mediating these bone-specific effects, we performed histologic evaluation of the tibiale (limited erosions) compared to the cuboid and talus (severe erosions) in 5-month-old females. H&E-OG staining revealed that the bones with severe erosions exhibit significantly increased synovial area surrounding the bone compared to the tibiale in TNF-Tg mice (Fig 4G–4K; TNF-Tg cuboid $1.78\pm0.76$ vs tibiale $0.85\pm0.66$ synovial/tissue area, *p<0.0001*). However, the increased synovitis was not consistently associated with a change in osteoclast number for the cuboid or talus vs tibiale, which may represent a

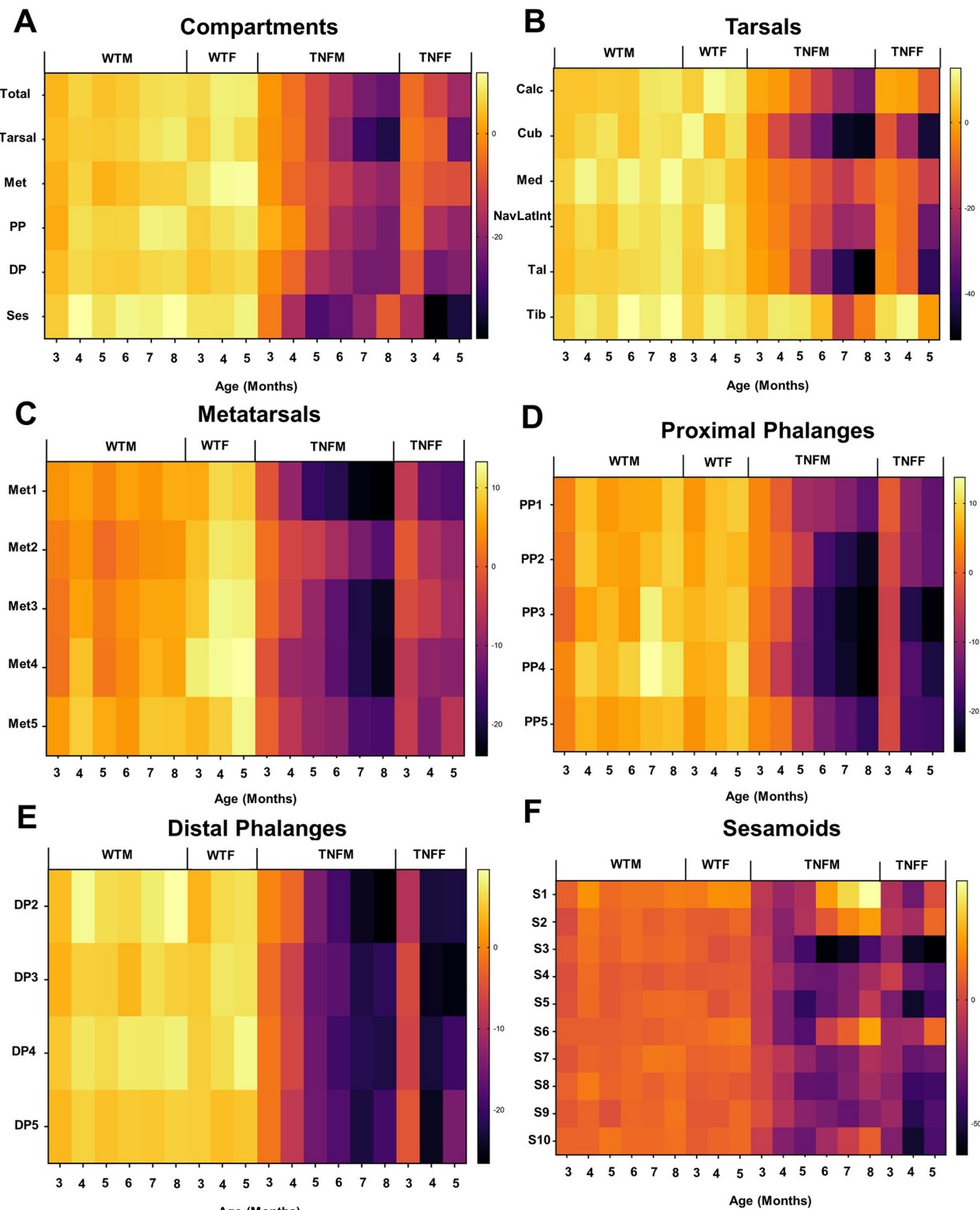

**Fig 3. Erosive arthritis severity and progression occurs in a bone-dependent and sexually dimorphic manner.** To visualize the bone-specific effects within the compartments of the hindpaw (**A**), we plotted the mean percent change from 2-month baseline bone volumes for each individual bone as heatmaps (**B-F**). Note the accelerated onset of bone erosions in female mice across each compartment of the hindpaw, along with the variable arthritic progression between bones, most notable in the tarsal region (**B**). Representative micro-CT images of the bones within each compartment across time are provided in S3–S9 Figs.

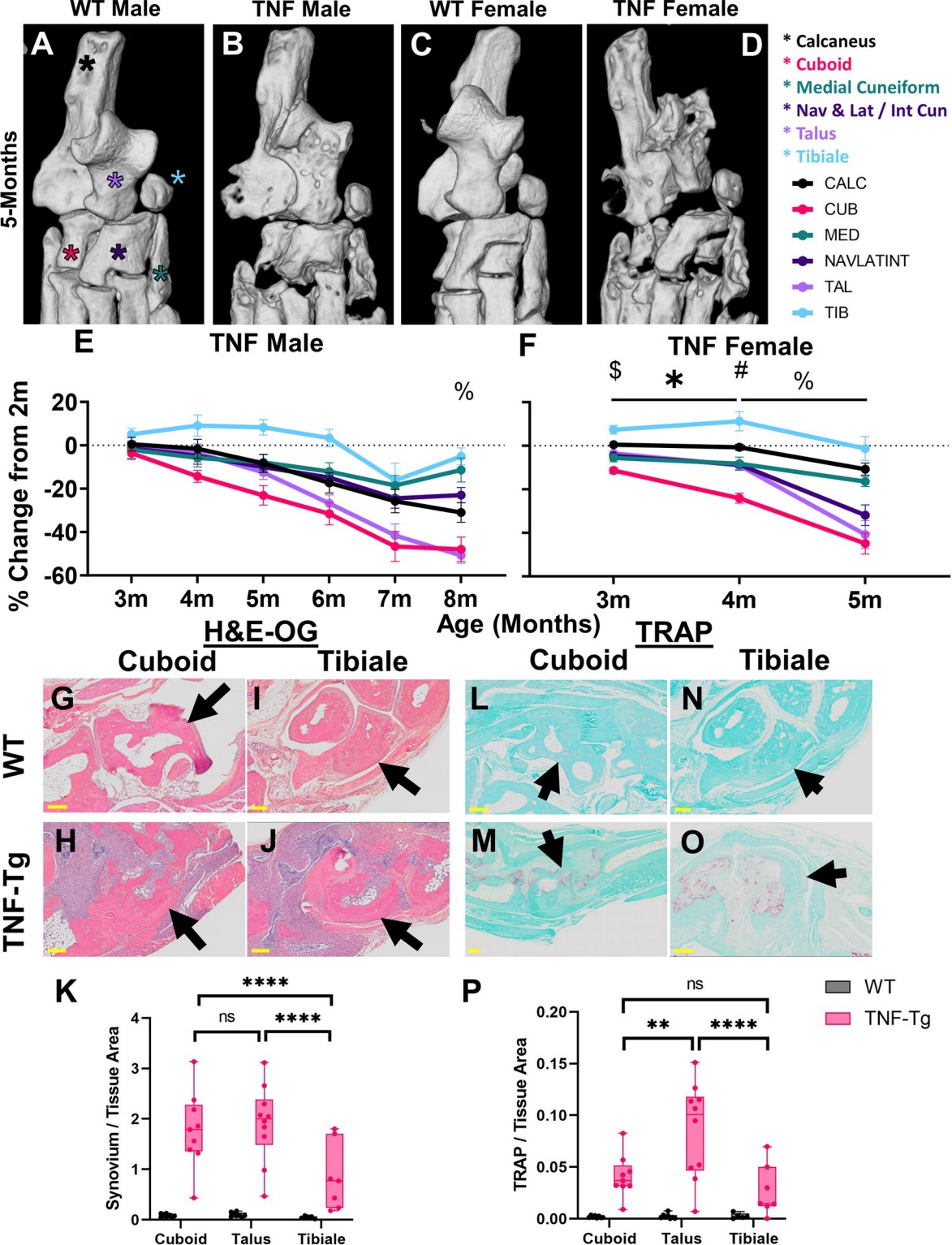

**Fig 4. The cuboid is an early biomarker of erosive arthritis in TNF-Tg female mice.** To assess the specific bones driving the dynamic temporal changes in the tarsals of TNF-Tg mice, we compared bone loss between individual bones. High-magnification images of the tarsal compartment at 5-months of age, with the individual bones indicated by stars (calcaneus, black; cuboid, pink; medial cuneiform, green; fused navicular, lateral cuneiform and intermediate cuneiform, dark purple; talus, light purple; and tibiale, blue), are shown for each group **(A-D)**. TNF-Tg males exhibited no significant difference in erosions between individual bones in the tarsal region, except for the tibiale

with relatively higher bone volumes (% *p<0.05*, vs all bones besides medial cuneiform) **(E)**. On the other hand, TNF-Tg females showed early erosive activity localized specifically to the cuboid bone with significantly reduced bone volumes compared to the other bones in the tarsal region by 3- and 4-months of age ($ cuboid vs all bones except medial cuneiform, # cuboid vs all bones, *p<0.05*). In addition, significant decreases in bone volume over time from 3- to 4-months of age were noted only in the cuboid and NAVLATINT (* *p<0.05*). All bones, except for the tibiale, showed significantly reduced bone volumes from 4- to 5-months of age (% *p<0.05*), explaining the time-dependent erosions of the tarsal compartment during this timeframe, noted in Fig 2 **(F)**. Histologic evaluation of the cuboid or talus (erosion susceptible) and tibiale (erosion resistant) revealed increased synovitis (H&E-OG) surrounding the cuboid **(O-K)**, but with inconsistent changes in number of osteoclasts (TRAP) compared to the tibiale in 5-month-old TNF-Tg females **(L-P)**. Statistics: 2-way ANOVA (males and histology) and mixed effects analysis (females) with Tukey's multiple comparisons **(E-F, K and P)**; ** *p<0.01*, *** *p<0.0001*.

ceiling of erosive activity for the cuboid at 5-months (Fig 4L–4P; TNF-Tg cuboid 0.041 ±0.020 vs tibiale 0.027±0.025 TRAP/tissue area, *p>0.05*). Together, these findings suggest that particular bones serve as time- and sex-dependent biomarkers to evaluate disease progression in TNF-Tg mice, where the cuboid volume was identified as an ideal bone metric to longitudinally monitor early erosive activity and synovitis in females with inflammatory arthritis. In males, the cuboid similarly exhibits the most profound volume changes that remain consistent across disease severity, while the talus shows comparable findings at later ages (i.e., 6–8 months), albeit without statistically significant findings compared to other tarsal bones.

## Anterolateral tarsal bones show female-specific early onset erosions in TNF-Tg mice

Based on the differential progression of erosions between male and female TNF-Tg mice, and the well-established sexual dimorphisms within this mouse model [44], we directly compared percent change in bone volume between the sexes. When normalized to relative bone size between the groups, changes in total hindpaw volume show no changes between male and female TNF-Tg mice (Fig 5A; 5-months TNF-Tg: males -10.3±6.5% vs females -17.6±3.7%, *p>0.05*). However, TNF-Tg females demonstrated a distinct decline in tarsal bones by 5-months of age compared to males (Fig 5B; 5-months TNF-Tg: males -10.3±9.5% vs females -24.9±10.9%, *p<0.05*). Interestingly, specific bones in the tarsal region, such as the calcaneus (5-months TNF-Tg: males -8.3±11.6% vs females -10.8±8.8%), medial cuneiform (males -7.9 ±10.1% vs females -16.5±7.3%), and tibiale (males 8.3±10.0% vs females -1.3±17.4%), exhibited no sexual dimorphisms in bone erosions across time (Fig 5C and S10A and S10B Fig; monomorphic, *p>0.05* TNF-Tg males vs females). On the other hand, particular bones, such as the talus (males -12.5±9.4% vs females -40.8±20.3%), cuboid (males -23.1±12.7 vs females -44.9 ±15.0%), and NAVLATINT (males -10.1±7.0% vs females -31.9±15.0%), showed notable sex-dependent erosive activity with increased severity in TNF-Tg females compared to males by 5-months of age (Fig 5D and S10C and S10D Fig; dimorphic, *p<0.05* TNF-Tg males vs females at 5-months). Histologic assessment of the talus (dimorphic) between male and female mice further revealed that TNF-Tg females exhibit both increased synovitis (males 0.52±0.030 vs females 1.91±0.76 synovial/tissue area, *p<0.0001*) and osteoclasts (males 0.016±0.002 vs females 0.090±0.053 TRAP/tissue area, *p<0.01*) in the region of the talus compared to their male counterparts (Fig 5E–5N). To visualize the localization of the monomorphic (yellow) and dimorphic (red) erosive bones within the TNF-Tg ankles, representative images of the tarsal regions at 5-months of age for male and female mice are shown (reproduced from Fig 4B, 4D and S5 Fig for emphasis). Notably, the bones that exhibit sexually dimorphic erosions localize specifically to the anterolateral region of the ankle, which suggests that this region is more susceptible to arthritic progression (Fig 5O and 5P).

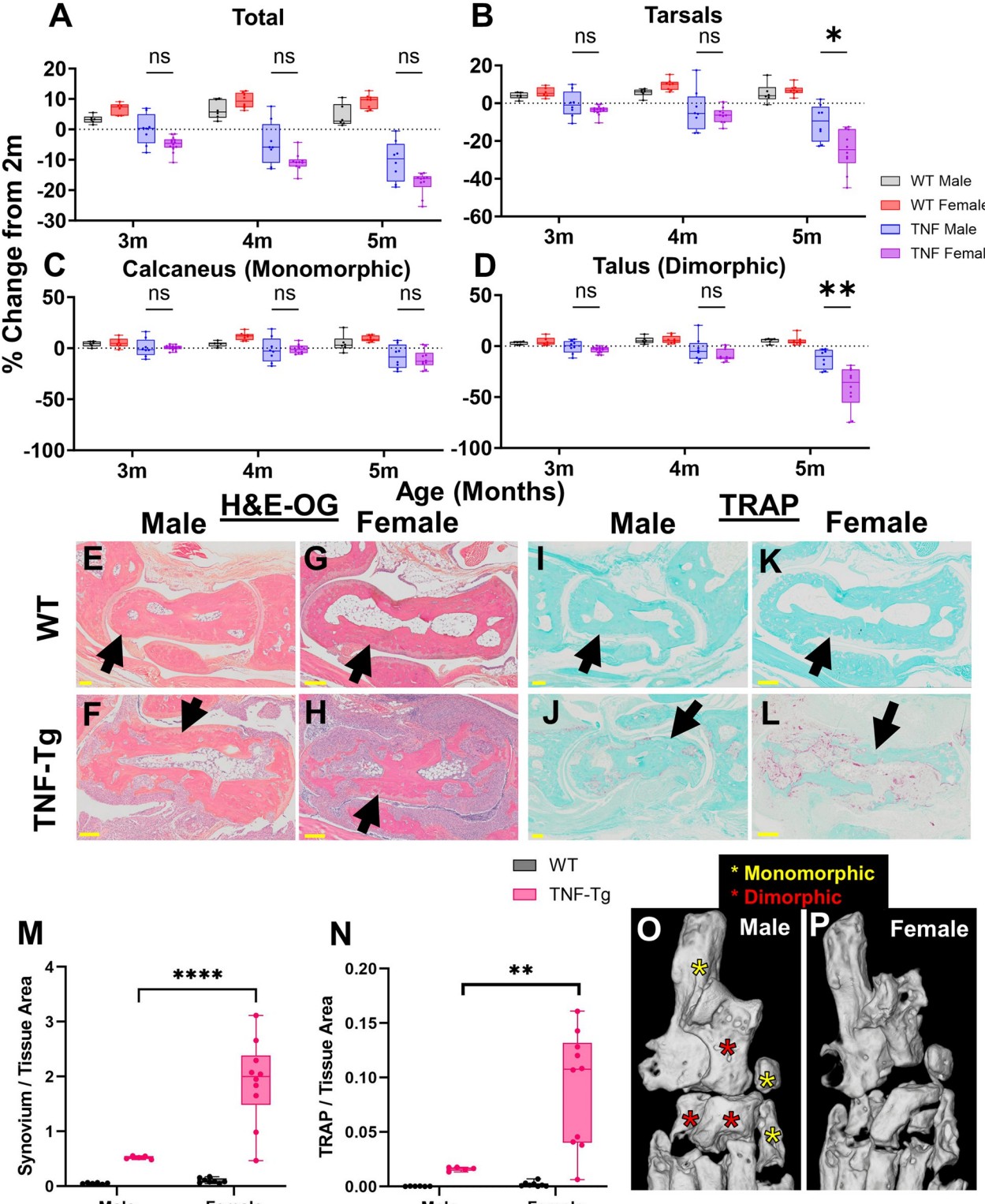

**Fig 5. Anterolateral tarsal bones show female-specific early onset erosions in TNF-Tg mice.** Sex-dependent differences in bone erosions within the tarsal compartment were evaluated. While total bone volume showed no significant changes between TNF-Tg males and females (**A**), the tarsal compartment exhibited reduced bone volumes in females compared to males at 5-months of age (**B**). Interestingly, for bones such as the calcaneus (**C**, along with medial cuneiform and tibiale in S10 Fig), there were no changes noted between the sexes. The significant reduction in tarsal bone volume in females was instead driven by sex-specific erosive activity in bones such as the talus (**D**, along with cuboid and fused navicular, lateral

cuneiform and intermediate cuneiform in S10 Fig). Histologic evaluation in male versus female TNF-Tg mice at 5-months of age revealed both increased synovitis and number of osteoclasts in TNF-Tg females compared to males (**E-N**). Interestingly, the bones associated with erosion of the tarsals in females are notably localized to the anterolateral region of the ankle (red stars), while those without sexual dimorphism are oriented posteromedially (yellow stars). Representative micro-CT images of male (**O**) and female (**P**) TNF-Tg mice at 5-months of age (reproduced from Figs 4B and 4D and S5 Fig for convenience) exhibit the lateralization of the sexually dimorphic effects. Statistics: Mixed effects analysis (**A-D**) and 2-way ANOVA (**M-N**) with Tukey's multiple comparisons; * $p<0.05$, ** $p<0.01$, **** $p<0.0001$.

## Specific bones exhibit differential responses to anti-TNF therapy in TNF-Tg mice

After characterizing the natural history of erosive activity, we next assessed the differential capacity for specific bones to recover with anti-TNF therapy towards defining biomarkers to evaluate pharmaceutical treatments efficiently and accurately in the TNF-Tg preclinical model of arthritis. Male WT (vehicle treated) and placebo or anti-TNF treated TNF-Tg mice were evaluated by micro-CT every 3-weeks during a 6-week treatment period [50], and representative images are shown at 6-weeks post-treatment (wpt). Note the severe erosions in the placebo-treated ankles relative to the bone recovery with anti-TNF therapy, albeit with retention of bone shape deformities compared to WT (Fig 6A–6C). To demonstrate the effectiveness of anti-TNF therapy, the total tarsal compartment showed significantly increased bone volumes in anti-TNF versus placebo by 3wpt, while anti-TNF remained at lower volumes relative to WT out to 6wpt (Fig 6D; 6wpt: WT 6.6±0.24 vs placebo 3.4±0.50 vs anti-TNF 4.8±0.23 mm$^3$, $p<0.05$). Certain bones, such as the tibiale (3wpt: placebo 0.16±0.043 vs anti-TNF 0.22±0.034 mm$^3$), medial cuneiform (placebo 0.33±0.042 vs anti-TNF 0.39±0.032 mm$^3$), talus (placebo 0.53±0.17 vs anti-TNF 0.81±0.094 mm$^3$), and calcaneus (placebo 1.6±0.20 vs anti-TNF 1.8 ±0.11 mm$^3$, exhibited significantly increased bone volumes with anti-TNF compared to placebo by 3wpt ($p<0.05$), while other bones showed delayed bone recovery until 6wpt, such as the cuboid (3wpt: placebo 2.6±0.074 vs anti-TNF 0.31±0.071 mm$^3$) and NAVLATINT (placebo 0.78±0.10 vs anti-TNF 0.89±0.11 mm$^3$; $p>0.05$). Note that specific bones demonstrated a contribution of increased volume beyond the benefits of anti-TNF therapy (i.e., tibiale with increases in placebo bone volume across time). In addition, certain bones showed increased treatment effect with limited variance between samples compared to others (i.e., talus & calcaneus vs NAVLATINT) (Fig 6E–6J). Thus, to demonstrate the bones with the greatest biomarker potential with treatment, we quantitatively compared the contribution of treatment and time as the effect size between anti-TNF and placebo-treated samples through calculation of $\eta^2$ (sample association within study). Through this approach, the talus ($\eta^2 = 0.21$) and calcaneus ($\eta^2 = 0.22$) were differentiated as bones that exhibited the greatest effect sizes (large effect: $\eta^2 > 0.138$, dashed line) with treatment across time associated with rapid (by 3wpt) and dramatic increases in bone volume with anti-TNF therapy compared to placebo (Fig 6K). To validate these findings, we similarly quantified comparable metrics of effect size, partial $\eta^2$ and $\omega^2$, which each provide different considerations for error within groups, and consistently identified the talus (partial $\eta^2 = 0.20$, $\omega^2 = 0.21$) and calcaneus (partial $\eta^2 = 0.19$, $\omega^2 = 0.21$) as the bones with the greatest and large effect (>0.138; equations provided in Materials and methods, Statistics; S11 Fig). Thus, the talus and calcaneus serve as specific and reliable biomarkers to evaluate treatment response of erosive arthritis in TNF-Tg mice.

## Discussion

Through advances in high-throughput micro-CT analysis techniques, we demonstrated the capacity to perform a comprehensive assessment of bone-specific imaging biomarkers with longitudinal monitoring of erosions in arthritic conditions affecting complex joints. Such a

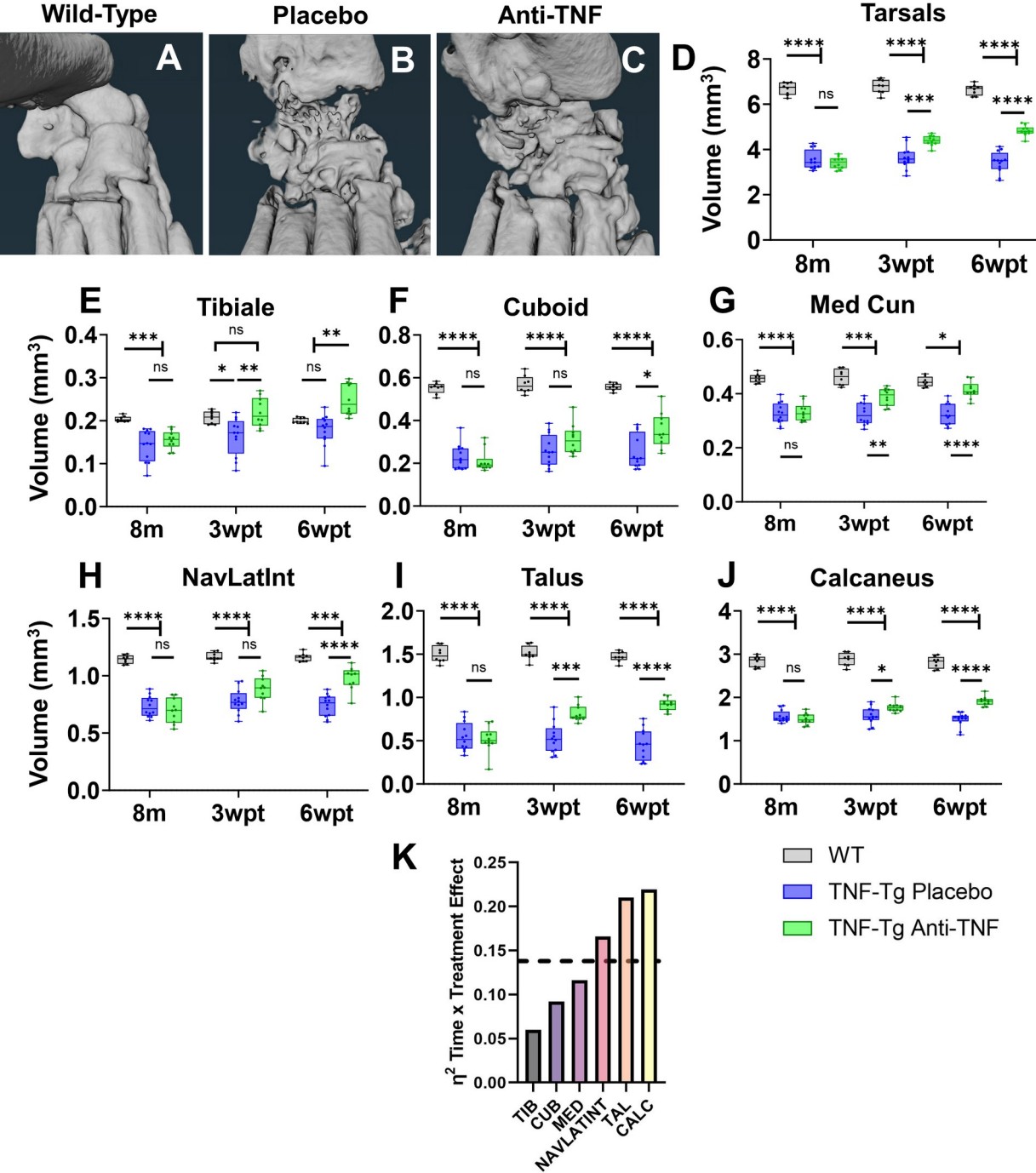

**Fig 6. Specific bones exhibit differential responses to anti-TNF therapy in TNF-Tg mice.** As we have noted that 6-weeks of anti-TNF therapy is sufficient to recover talus bone volumes in TNF-Tg male mice with severe arthritis at 8-months of age [50], we evaluated the response of individual bones in the tarsal region to determine reliable biomarkers of treatment effectiveness. Representative micro-CT images of the tarsal region in wild-type, placebo (IgG1 isotype antibodies) treated TNF-Tg, and anti-TNF treated TNF-Tg mice are shown. Note the general increase in bone volumes with anti-TNF therapy, although with continued abnormal bone morphology relative to wild-type (**A-C**). Measurements of total tarsal bone volume indicated effective response to anti-TNF relative to placebo treatment by 3-weeks post-treatment (wpt), although bone volumes remained lower than wild-type at 6wpt (**D**). Direct comparison of the treatment response between placebo and anti-TNF therapy for the individual bones revealed a heterogenous response. Although all bones exhibited a significantly increased bone volumes in anti-TNF compared to placebo treated TNF-Tg mice by 6wpt (**E-J**), effects of treatment type over time exhibited differential response between bones, quantified by $\eta^2$ effect size between placebo and anti-TNF groups (**K**, dashed line = large effect, >0.138). Statistics: 2-way ANOVA with Tukey's multiple comparisons (**D-J**); * $p<0.05$, ** $p<0.01$, *** $p<0.001$, **** $p<0.0001$. Sample sizes: wild-type, n = 8; placebo, n = 12; anti-TNF, n = 10 hindpaws (**D-J**).

rigorous approach thus catalogues a natural history with clear rationales for analysis of particular bones in future studies to ensure appropriate interpretations, and also provides evidence for further investigation to understand the localization of erosive pathology. Similar to clinical disease that often first identifies periarticular erosions in peripheral joints of the phalanges [59], we discovered that the phalange-associated bones consistently exhibited the earliest arthritic progression in TNF-Tg mice. The deterioration of these distal bones was then shortly followed by a rapid and dramatic decrease in bone volume in the proximal tarsal bones of the ankle joint, most notable in female mice. However, we discovered that each tarsal bone showed unique temporal progression of disease, where the cuboid specifically demonstrated early-onset and sustained erosions across time in females. Of note, the cuboid exhibited a similar trend with early and consistent bone volume decline in males, but with less dramatic effect and was not statistically different compared to other tarsals. Thus, the cuboid served as a consistent biomarker of erosive arthritis throughout both initial (primarily phalanges) and late (primarily other tarsals) manifestations of the disease, unlike all other identified bones where rates in bone volume change were temporally dependent. Most notable was the talus, which showed minimal erosions early in disease progression and more prominent bone volume decline at later stages, by 5-months in females and 6-8-months in males, albeit not statistically different compared to other tarsals.

The distinct mechanisms that mediate cuboid-selective erosive activity remain unclear, but appear to be associated with the particular anatomic localization within the ankle joint. Notably, the most anterolateral bones, such as the cuboid, talus, and NAVLATINT, were identified as the tarsals with the greatest reduction in bone volumes over time, and these effects were exacerbated in female mice. The anterolateral specification of these bones may be related to an environment more susceptible to inflammation and biomechanical strain related to traversing tendons. For instance, the lateral aspect of the hindpaw contains a greater number of close proximity tendons compared to the medial side [47], and thus association with adjacent tenosynovitis may drive bone-specific differences. An additional possibility is the lateral hindpaw experiences greater biomechanical strain related to differential pressure loading, which ought to be investigated via gait analysis in future studies. In fact, prior gait analysis demonstrated that female mice exhibit decreased stride length and velocity along with increased cadence compared to males [60], suggesting that the sexual dimorphism in bone erosions may in part be related to relatively increased step count for a particular distance. We also recently showed that, given the uninhibited opportunity for exercise via wheel running, WT females exhibit increased total distance of activity compared to male counterparts [61]. Further investigation into the effects of exercise on bone-specific erosions, and the potential impact of non-pharmaceutical hindpaw stabilization strategies to limit detrimental effects on the anterolateral hindpaw, are warranted.

Regardless of the initiating events, our data provides evidence that the bones exhibiting increased erosions are directly related to enhanced surrounding synovitis and osteoclast number. Although we also made efforts to evaluate an osteoblast phenotype, as osteoblasts are known to exhibit dysfunction in TNF-Tg mice [62], options for feasible high-throughput analysis of osteoblasts by standard histologic stains remain limited. In TNF-Tg histology, the identification of osteoblasts by traditional toluidine blue staining [63] exhibits remarkable difficulty as the dramatic synovitis and pannus invasion blurs the capacity for accurate osteoblast quantification. Additionally, the interpretation of identified osteoblasts could either represent recovery of bone or a response to surrounding inflammation generating peri-articular osteophytes that are non-contributory to effective joint repair. While recent advances have been made in bone histology to highlight osteoid compared to calcified bone, even in decalcified tissue used for histology [64], to our knowledge there are no stains specific for osteoid to

allow for detailed histomorphometric quantification of osteoblasts. Certain markers for immunohistochemistry, such as osteocalcin [65], can also be utilized to quantify osteoblasts, but exhibit limited feasibility in experimental designs necessitating large sample sizes due to reduced throughput and thus increased possibility of batch effects. Further experimentation towards developing improved high-throughput and specific osteoblast staining protocols would provide considerable benefit to bone research.

Our study has additional notable limitations, including the specificity of our findings to the TNF-Tg mice and particular timeframe of anti-TNF treatment regimen. It remains possible that alternative disease models will exhibit distinct temporal bone volume changes across time, where a similar approach demonstrated in this work could rigorously elucidate the appropriate biomarkers for tracking bone pathology. While our work was focused on evaluating bone loss, there were also interesting dynamics in the bones that retained their volume, and even showed periods of potential bone formation without intervention after erosions (i.e., sesamoids (Fig 2J), tibiale, and medial cuneiform (Fig 4E)). However, this study was limited in the potential to investigate such mechanisms, which could be related to multiple possibilities, including temporal changes in bone remodeling or expansion of adjacent hyperdense tissue (i.e., fibrosis) that may have been detected as bone in threshold-reliant micro-CT analysis. Paired and targeted histology at these times of bone growth in future studies are needed for accurate interpretation of these unexplained volume trajectories. In addition, the findings related to treatment responses are limited to male mice with severe disease on anti-TNF therapy, where earlier intervention, different therapeutic options, or investigation in females may exhibit differential treatment effects. The quantification of bone growth with treatment ought to also be considered, where our approach solely evaluated change in bone volume, while particular localization of osteogenesis may be essential. For instance, interpretation of effective therapy may require consideration regarding the restoration of original bone shape and structure to promote appropriate function, as bone growth itself does not necessarily equate to clinical improvement. Advancements in such analysis strategies are warranted, and provide the potential for imaging to further predict clinically relevant and ideal treatment responses.

Importantly, the effective utilization of the semi-automated segmentation model [46] in both healthy and arthritic hindpaws offers opportunities for continued improvement and enhanced throughput of these analysis methods. For instance, recent advancements have been developed in deep learning models to perform fully automated segmentation of complex structures, such as the tibia or subchondral bone therein [66–69]. Through the segmentation of approximately 9,000 individual bones in this work, we have thus developed a remarkable resource of datasets that can be utilized to train completely automated deep learning models for further increased accuracy and efficiency. As noted above, these methods could then be directly implemented in other pre-clinical murine models, and these efforts offer a blueprint for adoption of similar techniques for detailed clinical investigations.

Overall, we demonstrated that bone-specific biomarkers of arthritic disease can be elucidated through innovations in high-throughput analysis of CT datasets. Further utilization of similar approaches would provide tremendous benefit to other pre-clinical arthritis models and has potential to enhance clinical diagnostics and detailed disease monitoring. Through high-resolution quantification of longitudinal bone erosions, these approaches provide an opportunity to identify early patterns of arthritic progression as biomarkers to guide timely therapeutic intervention. These strategies are also essential for experimental evaluation in order to provide a comprehensive analytical approach of complex joints that allows for a rigorous rationale to correctly interpret erosive activity or additional outcomes of interest throughout future investigations.

## Conclusions

We demonstrated the capacity to perform comprehensive and bone-specific analysis of arthritic erosions in complex joints through recent advances in high-throughput and automated segmentation approaches. These methods identified the cuboid as a reliable, early, and sexually-dimorphic biomarker of inflammatory-erosive arthritis in the TNF-Tg pre-clinical model of RA, which provides a rigorous rationale as an outcome measure in future studies. Further efforts are also warranted to elucidate a greater understanding of the cellular and/or biomechanical mechanisms driving the bone-specific effects to guide development of novel therapeutics. Together, this work highlights the utility in strategically utilizing CT imaging to determine sensitive and quantitative biomarkers of erosion in future investigation towards early treatment intervention and expedited evaluation of response to therapy in both pre-clinical and clinical studies.

## Permissions for reuse

Datasets associated with wild-type endpoints in Figs 1–5 are derived from the datasets utilized in our previous work describing the development of the hindpaw segmentation model: Kenney H.M., Peng Y., Chen K.L. et al. A high-throughput semi-automated bone segmentation workflow for murine hindpaw micro-CT datasets. Bone Rep 16, 101167 (2022). https://doi.org/10.1016/j.bonr.2022.101167 [46]. Reuse of this material is protected by the Creative Commons Attribution-NonCommercial-NoDerivatives 4.0 International License https://creativecommons.org/licenses/by-nc-nd/4.0/legalcode. As authors of the referenced work, we retain the right to prepare other derivative works via Author's Rights from Elsevier https://beta.elsevier.com/about/policies-and-standards/copyright. The datapoints have been revisualized for comparison with other bones or TNF-Tg mice with inflammatory-erosive arthritis as outcomes relevant to the current work.

Datapoints associated with talus bone volumes at ages 2-5-months-old in Figs 2–5 were previously published in Kenney, H.M., Wood, R.W., Ramirez, G. et al. Implementation of automated behavior metrics to evaluate voluntary wheel running effects on inflammatory-erosive arthritis and interstitial lung disease in TNF-Tg mice. Arthritis Res Ther 25, 17 (2023). https://doi.org/10.1186/s13075-022-02985-6 [61]. Permission for reuse is provided under the Creative Commons Attribution 4.0 International License https://creativecommons.org/licenses/by/4.0/legalcode. The datapoints are adapted to be included within the tarsal region of the hindpaw or revisualized with plots relevant for comparison in the current work.

Datapoints for talus bone volumes with anti-TNF treatment in Fig 6 were previously published in Kenney, H.M., Peng, Y., Bell, R.D. et al. Persistent popliteal lymphatic muscle cell coverage defects despite amelioration of arthritis and recovery of popliteal lymphatic vessel function in TNF-Tg mice following anti-TNF therapy. Sci Rep 12, 12751 (2022). https://doi.org/10.1038/s41598-022-16884-y [50]. Permission for reuse is provided under the Creative Commons Attribution 4.0 International License https://creativecommons.org/licenses/by/4.0/legalcode. Additional datapoints are used in this work corresponding to mice that were evaluated at the same timepoints, but not associated with endpoints in the previously published study. The datapoints are adapted for revisualization in plots relevant for comparison in the current work.

## Supporting information

**S1 Fig. Histologic identification of specific tarsal bones.** Representative H&E-OG-stained histologic sections of the tarsal region in a wild-type ankle is provided to demonstrate the identification of the talus (**A**, purple asterisk), cuboid (**B**, pink asterisk), and tibiale (**C**, blue

asterisk) relative to closely articulating bones. A micro-CT image is provided with specific tarsal bones highlighted by color coded asterisks (**D**, reproduced from Fig 4A) to directly visualize the sectioning planes (white dashed lines) utilized to evaluate these particular bones by histology.
(TIF)

**S2 Fig. Histomorphometric segmentation of synovial and osteoclast areas.** Representative histologic sections of a talus stained with H&E-OG (**A**) and TRAP (**B**) from WT (left) and TNF-Tg (right) cohorts with demonstration of segmentation for quantification of corresponding areas using the Visiopharm software are provided. For the H&E-OG, the original staining is shown (top) with the resultant segmentation (bottom; green = synovium, blue = bone and soft tissue, red = adipose and background) adjacent to a corresponding transparent overlay of the staining and segmentation (middle) (**A**). Similarly, the TRAP staining (blue = bone and soft tissue, red = TRAP) is segmented with a bright red overlay, and the pink dashed line represents the region of interest surrounding the talus where the analysis was performed (**B**).
(TIF)

**S3 Fig. Micro-CT images of the dorsal hindpaw across time.** A representative 3D rendering of micro-CT datasets for WT (**A**, tarsals = red asterisk, metatarsals = blue asterisk, proximal phalanges = yellow asterisk, distal phalanges = white asterisk) and TNF-Tg (**B**) male dorsal hindpaws from 2–8 months of age at monthly intervals are provided. Similar images for WT (**C**) and TNF-Tg (**D**) female dorsal hindpaws from 2–5 months of age are shown.
(TIF)

**S4 Fig. Micro-CT images of the plantar hindpaw across time.** A representative 3D rendering of micro-CT datasets for WT (**A**, tarsals = red asterisk, metatarsals = blue asterisk, proximal phalanges = yellow asterisk, distal phalanges = white asterisk, sesamoids = pink asterisk) and TNF-Tg (**B**) male plantar hindpaws from 2–8 months of age at monthly intervals are provided. Similar images for WT (**C**) and TNF-Tg (**D**) female plantar hindpaws from 2–5 months of age are shown.
(TIF)

**S5 Fig. Micro-CT images of the tarsal compartment across time.** A representative 3D rendering of micro-CT datasets for WT (**A**, calcaneus = red asterisk, cuboid = blue asterisk, navicular & lateral cuneiform / intermediate cuneiform [variably fused] = yellow asterisk, medial cuneiform = white asterisk, talus = pink asterisk, and tibiale = green asterisk) and TNF-Tg (**B**) male tarsal regions from 2–8 months of age at monthly intervals are provided. Similar images for WT (**C**) and TNF-Tg (**D**) female tarsal compartments from 2–5 months of age are shown.
(TIF)

**S6 Fig. Micro-CT images of the metatarsal compartment across time.** A representative 3D rendering of micro-CT datasets for WT (**A**, $1^{st}$ metatarsal = red asterisk, $2^{nd}$ metatarsal = blue asterisk, $3^{rd}$ metatarsal = yellow asterisk, $4^{th}$ metatarsal = white asterisk, $5^{th}$ metatarsal = pink asterisk) and TNF-Tg (**B**) male metatarsal regions from 2–8 months of age at monthly intervals are provided. Similar images for WT (**C**) and TNF-Tg (**D**) female metatarsal compartments from 2–5 months of age are shown.
(TIF)

**S7 Fig. Micro-CT images of the proximal phalange compartment across time.** A representative 3D rendering of micro-CT datasets for WT (**A**, $1^{st}$ proximal phalange = red asterisk, $2^{nd}$ proximal phalange = blue asterisk, $3^{rd}$ proximal phalange = yellow asterisk, $4^{th}$ proximal phalange = white asterisk, $5^{th}$ proximal phalange = pink asterisk) and TNF-Tg (**B**) male

proximal phalange regions from 2–8 months of age at monthly intervals are provided. Similar images for WT **(C)** and TNF-Tg **(D)** female proximal phalange compartments from 2–5 months of age are shown. Bones are numbered 1–5 based on digits from medial to lateral.
(TIF)

**S8 Fig. Micro-CT images of the distal phalange compartment across time.** A representative 3D rendering of micro-CT datasets for WT **(A**, 2nd distal phalange = red asterisk, 3rd distal phalange = blue asterisk, 4th distal phalange = yellow asterisk, 5th distal phalange = white asterisk**)** and TNF-Tg **(B)** male distal phalange regions from 2–8 months of age at monthly intervals are provided. Similar images for WT **(C)** and TNF-Tg **(D)** female distal phalange compartments from 2–5 months of age are shown. Bones are numbered 2–5 based on digits from medial to lateral; note the 1st digit does not have a distal phalange.
(TIF)

**S9 Fig. Micro-CT images of the sesamoid compartment across time.** A representative 3D rendering of micro-CT datasets for WT **(A**, sesamoids 1 & 2 = red asterisk, sesamoids 3 & 4 = blue asterisk, sesamoids 5 & 6 = yellow asterisk, sesamoids 7 & 8 = white asterisk, sesamoids 9 & 10 = pink asterisk**)** and TNF-Tg **(B)** male sesamoid regions from 2–8 months of age at monthly intervals are provided. Similar images for WT **(C)** and TNF-Tg **(D)** female sesamoid compartments from 2–5 months of age are shown. Bones are numbered 1–10 from medial to lateral with 2 sesamoids per digit.
(TIF)

**S10 Fig. Identification of tarsal bones with and without sexually dimorphic progression of erosive arthritis.** Corresponding with the sexually monomorphic (yellow asterisks) and dimorphic (red asterisks) bones highlighted in Fig 5O and 5P, quantification of normalized bone volumes across time for the medial cuneiform **(A)** and tibiale **(B)** (monomorphic) along with the cuboid **(C)** and NAVLATINT **(D)** (dimorphic) are provided. Statistics: Mixed effects analysis with Tukey's multiple comparisons **(A-D)**; * $p<0.05$, ** $p<0.01$.
(TIF)

**S11 Fig. Effect sizes of anti-TNF treatment across time for specific tarsal bones.** Associated with the quantification of effect size by eta-squared ($\eta^2$) shown in Fig 6K, we further evaluated whether the same relationships remained consistent across alternative methods for effect size assessment, including partial $\eta^2$ **(A)** and omega-squared ($\omega^2$) **(B)** (equations provided in Materials and methods, Statistics). Consistent with evaluation of $\eta^2$ both partial $\eta^2$ and $\omega^2$ identified the talus (partial $\eta^2 = 0.20$, $\omega^2 = 0.21$) and calcaneus (partial $\eta^2 = 0.19$, $\omega^2 = 0.21$) as the bones with the greatest effect sizes (large effect size $>0.138$, dashed black lines).
(TIF)

## Acknowledgments

We would like to thank Jannsen (J&J) for providing the anti-TNF and placebo IgG1 monoclonal antibodies used in this study. We would also like to thank the faculty and staff of the Histology, Biochemistry, and Molecular Imaging core and the Biomechanics, Biomaterials, and Multimodal Tissue Imaging core at the University of Rochester Medical Center for their contributions to this work.

## Author Contributions

**Conceptualization:** H. Mark Kenney, Ronald W. Wood, Edward M. Schwarz, Hani A. Awad.

**Data curation:** H. Mark Kenney, Kiana L. Chen, Lindsay Schnur, Jeffrey I. Fox.

**Formal analysis:** H. Mark Kenney.

**Funding acquisition:** H. Mark Kenney.

**Investigation:** H. Mark Kenney.

**Methodology:** H. Mark Kenney.

**Resources:** Edward M. Schwarz.

**Software:** H. Mark Kenney, Ronald W. Wood.

**Supervision:** Lianping Xing, Christopher T. Ritchlin, Homaira Rahimi, Edward M. Schwarz, Hani A. Awad.

**Visualization:** H. Mark Kenney.

**Writing – original draft:** H. Mark Kenney.

**Writing – review & editing:** Kiana L. Chen, Lindsay Schnur, Jeffrey I. Fox, Ronald W. Wood, Lianping Xing, Christopher T. Ritchlin, Homaira Rahimi, Edward M. Schwarz, Hani A. Awad.

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
