## [Decision Letter · Decision Letter 0]

4 Apr 2024

PONE-D-24-06132High-Throughput Micro-CT Analysis Identifies Sex-Dependent Biomarkers of Erosive Arthritis in TNF-Tg Mice and Differential Response to Anti-TNF TherapyPLOS ONE

Dear Dr. Awad,

Thank you for submitting your manuscript to PLOS ONE. After careful consideration, we feel that it has merit but does not fully meet PLOS ONE’s publication criteria as it currently stands. Therefore, we invite you to submit a revised version of the manuscript that addresses the points raised during the review process.

We look forward to receiving your revised manuscript.

Kind regards,

Sadiq Umar

Academic Editor

PLOS ONE

Journal Requirements:

“We would like to thank Jannsen (J&J) for providing the anti-TNF and placebo IgG1 monoclonal antibodies used in this study. We would also like to thank the faculty and staff of the Histology, Biochemistry, and Molecular Imaging core and the Biomechanics, Biomaterials, and Multimodal Tissue Imaging core at the University of Rochester Medical Center for their contributions to this work.

Funding

This work was supported by funding from the National Institutes of Health (NIH): F30AG076326 (HMK), T32GM007356 (HMK), T32AR076950 (HMK, KLC), R01AR069000 (CTR), R01AR056702 (EMS), and P30AR069655. HMK is a trainee in the Medical Scientist Training Program funded by NIH T32GM007356. The content is solely the responsibility of the authors and does not necessarily represent the official views of the National Institution of General Medical Science or NIH. The funders did not play any role in the study design, data collection and analysis, decision to publish, or preparation of the manuscript.”

“This work was supported by funding from the National Institutes of Health (NIH): F30AG076326 (HMK), T32GM007356 (HMK), T32AR076950 (HMK, KLC), R01AR069000 (CTR), R01AR056702 (EMS), and P30AR069655. HMK is a trainee in the Medical Scientist Training Program funded by NIH T32GM007356. The content is solely the responsibility of the authors and does not necessarily represent the official views of the National Institution of General Medical Science or NIH. The funders did not play any role in the study design, data collection and analysis, decision to publish, or preparation of the manuscript.”

3. We noted in your submission details that a portion of your manuscript may have been presented or published elsewhere. [Bone Rep 16, 101167 (2022). https://doi.org/10.1016/j.bonr.2022.101167, rthritis Res Ther 25, 17 (2023). https://doi.org/10.1186/s13075-022-02985-6, and Sci Rep 12, 12751 (2022). https://doi.org/10.1038/s41598-022-16884-y] Please clarify whether this [conference proceeding or publication] was peer-reviewed and formally published. If this work was previously peer-reviewed and published, in the cover letter please provide the reason that this work does not constitute dual publication and should be included in the current manuscript.

4. In the online submission form you indicate that your data is not available for proprietary reasons and have provided a contact point for accessing this data. Please note that your current contact point is a co-author on this manuscript. According to our Data Policy, the contact point must not be an author on the manuscript and must be an institutional contact, ideally not an individual. Please revise your data statement to a non-author institutional point of contact, such as a data access or ethics committee, and send this to us via return email. Please also include contact information for the third party organization, and please include the full citation of where the data can be found.

Reviewers' comments:

Reviewer's Responses to Questions

**Comments to the Author**

1. Is the manuscript technically sound, and do the data support the conclusions?

Reviewer #1: Yes

2. Has the statistical analysis been performed appropriately and rigorously? 

Reviewer #1: Yes

3. Have the authors made all data underlying the findings in their manuscript fully available?

Reviewer #1: Yes

4. Is the manuscript presented in an intelligible fashion and written in standard English?

Reviewer #1: Yes

5. Review Comments to the Author

Reviewer #1: In the manuscript by Kenney et al, the authors have developed a semi-automated method to quantify bone erosion in murine models of arthritis. This work follows upon an earlier publication in which the method was presented to quantify bone erosions in normal, non-arthritic mice. The authors have provided significant supporting data validating the technique, as well as provided quantitative evidence that the erosion begins at the phalanges before proceeding to tarsal bones in this model and is more significant in females compared to males. The paper is extremely well-written and the figures clearly presented. The conclusions of the authors are well-supported by the data.

I only have one question for the authors: In Figure 4, the tibiale and medial cunieform show a curious unexpected increase in bone volume from 7 to 8 months. The authors mention that this represents that the “a relative resistance to erosions”. However, these bones show clear erosions from 5 to 7 m. Could there be another explanation? Perhaps, the tissue becomes excessively fibrotic and this dense tissue is detected as “bone” (above the threshold utilized to segment bone)? Do the authors have histological sections at this timepoint to evaluate this? If this is indeed the explanation, the authors should mention this as a limitation.

6. PLOS authors have the option to publish the peer review history of their article (what does this mean?). If published, this will include your full peer review and any attached files.

Reviewer #1: No

---

## [Author Response · Author response to Decision Letter 0]

14 May 2024

Point-By-Point Response to Reviewers

Editor

We are very pleased to learn that the Reviewers find our manuscript of interest. Based on these thoughtful reviews, we modified and improved our manuscript as outlined below in blue. In addition, we have addressed the provided details to ensure compliance with journal requirements as documented. We are grateful for the opportunity for revision and thank you for the consideration of our manuscript in PLOS One. 

--

Thank you for providing the useful documents with clear instructions on particular formatting requirements throughout the manuscript. We have incorporated the requested changes, including: changing title and headings to sentence case with appropriate font sizing for each level, listed each author affiliation individually, reformatted figure citations, modified reference citations to square brackets, and removed funding and competing interests from Acknowledgements (as also noted below). 

2. We note that you have provided funding information that is currently declared in your Funding Statement. However, funding information should not appear in the Acknowledgments section or other areas of your manuscript. We will only publish funding information present in the Funding Statement section of the online submission form. Please remove any funding-related text from the manuscript and let us know how you would like to update your Funding Statement. Please include your amended statements within your cover letter; we will change the online submission form on your behalf.

As noted above, the funding information (along with competing interest) has been removed from the Acknowledgements section. We will opt to retain the current version of the Funding Statement, which we will also note in the cover letter. 

3. We noted in your submission details that a portion of your manuscript may have been presented or published elsewhere. [Bone Rep 16, 101167 (2022). https://doi.org/10.1016/j.bonr.2022.101167, rthritis Res Ther 25, 17 (2023). https://doi.org/10.1186/s13075-022-02985-6, and Sci Rep 12, 12751 (2022). https://doi.org/10.1038/s41598-022-16884-y] Please clarify whether this [conference proceeding or publication] was peer-reviewed and formally published. If this work was previously peer-reviewed and published, in the cover letter please provide the reason that this work does not constitute dual publication and should be included in the current manuscript.

To provide clarification, all of the referenced works have been peer-reviewed and formally published. The utilization of the prior data does not constitute dual publication as the data has been repurposed for unique objectives, comparisons, and analyses. This information will also be provided in the cover letter, as requested.

4. In the online submission form you indicate that your data is not available for proprietary reasons and have provided a contact point for accessing this data. Please note that your current contact point is a co-author on this manuscript. According to our Data Policy, the contact point must not be an author on the manuscript and must be an institutional contact, ideally not an individual. Please revise your data statement to a non-author institutional point of contact, such as a data access or ethics committee, and send this to us via return email. Please also include contact information for the third party organization, and please include the full citation of where the data can be found.

Thank you for providing this important correction to ensure continuity with data availability for readers of this work. The manuscript has been revised to state: All data will be made available upon reasonable request through the University of Rochester’s Miner Library Data Services (miner_information@urmc.rochester.edu), per approved University data sharing policies.

References have been reviewed, and to our knowledge no cited work has been retracted.

Reviewer

In the manuscript by Kenney et al, the authors have developed a semi-automated method to quantify bone erosion in murine models of arthritis. This work follows upon an earlier publication in which the method was presented to quantify bone erosions in normal, non-arthritic mice. The authors have provided significant supporting data validating the technique, as well as provided quantitative evidence that the erosion begins at the phalanges before proceeding to tarsal bones in this model and is more significant in females compared to males. The paper is extremely well-written and the figures clearly presented. The conclusions of the authors are well-supported by the data.

I only have one question for the authors: In Figure 4, the tibiale and medial cunieform show a curious unexpected increase in bone volume from 7 to 8 months. The authors mention that this represents that the “a relative resistance to erosions”. However, these bones show clear erosions from 5 to 7 m. Could there be another explanation? Perhaps, the tissue becomes excessively fibrotic and this dense tissue is detected as “bone” (above the threshold utilized to segment bone)? Do the authors have histological sections at this timepoint to evaluate this? If this is indeed the explanation, the authors should mention this as a limitation.

We thank the Reviewer for the high praise of our work and appreciate the thoughtful comment on the erosive trajectory of specific bones to enhance the quality of the manuscript. As the Reviewer indicates, certain bones that retained the largest detected volumes at 8-months in males (i.e., tibiale and medial cuneiform) exhibited an initial period of erosions with notable increases in volume thereafter (Fig 4E). We also found this phenomenon intriguing and seemed to mirror a similar pattern that occurred for the sesamoid bones at earlier time periods, where there was a consistent increase in bone volume starting as early as 5-6 months to form the “U”-shape (Fig 2J). We acknowledge that our conclusion of “relative resistance to erosions” provided limited insights into the other possibilities to explain these unexpected changes in volume, including a shift to ossification in bone remodeling or expansion of dense fibrotic tissue. 

Although bone remodeling is overwhelmed by erosions during active rheumatoid arthritis, in the setting of disease remission bone growth may occur, but in an irregular and often non-physiologic manner [1]. It remains possible that joint and gait dynamics over time could alter susceptibility to biomechanical forces that promote the potential for erosion vs ossification, and thus certain bones may shift to be considered relatively “in remission”. This possibility is supported by the notable localization of the tibiale and medial cuneiform adjacent to each other on the medial aspect of the hindpaw, where both may experience similar changes in weight-bearing throughout disease progression. Moreover, it is notable that the bones with increased volume, such as the cuboid and sesamoids, exhibit a distinct ovoid shape compared to other bones in the hindpaw, and raises the possibility that particular bones may have optimal surface architecture for ossification to eventually balance or outcompete erosions. 

In addition, as the Reviewer keenly notes, we cannot rule out the potential that the increased volume was actually detecting hyperdense tissue (i.e., fibrosis) that fulfilled the threshold criteria of the micro-CT analysis, although was not truly bone at the tissue level. We regret to inform the Reviewer that we do not have available histological sections at the 8-month time period, as these animals continued as part of the anti-TNF portion of the study, where bone-targeted sectioning of the hindpaws was not performed [2]. To address this important point, we have added details to the Discussion on other possibilities beyond “erosion resistance” to explain the increased volume of certain bones, and indicated our limitation in histologic investigation that ought to be evaluated in future studies (text below). We also tempered the language in the Results to solely indicate such bones had higher volumes, where “relative resistance to erosions” is only one possibility.

Discussion:

“While our work was focused on evaluating bone loss, there were also interesting dynamics in the bones that retained their volume, and even showed periods of potential bone formation without intervention after erosions (i.e., sesamoids (Fig 2J), tibiale, and medial cuneiform (Fig 4E)). However, this study was limited in the potential to investigate such mechanisms, which could be related to multiple possibilities, including temporal changes in bone remodeling or expansion of adjacent hyperdense tissue (i.e., fibrosis) that may have been detected as bone in threshold-reliant micro-CT analysis. Paired and targeted histology at these times of bone growth in future studies are needed for accurate interpretation of these unexplained volume trajectories.”

Additional Comments

1. Figure 3 – timepoint indicators on heatmap were slightly adjusted for improved alignment and aesthetic.

2. Figure 4 – histology scale bars were noted to be inadvertently misplaced off of the images, which were corrected.

3. Minor edits were made to improve the writing along with corrections and clarification to sample sizes. Most notable was the error in the histology sample sizes listing n = 6 TNF-Tg males that were added for 5-6m histology, which should have instead indicated n = 6 males, split n = 3 WT and n = 3 TNF-Tg mice (n = 6 hindpaws each). We also added important details to explain the slightly decreased sample size of tibiale histology by n = 3 hindpaws vs cuboid or talus related to loss of tissue during optimization of the sectioning method. Lastly, we clarified that the unfortunate motion artifact in n = 1 WT male micro-CT dataset at 2-months of age also impacted downstream sample sizes that depended on this baseline measure, effectively providing only n = 6 as opposed to n = 8 datapoints for WT males. We apologize for the lack of these important details in the initial submission of the manuscript, and hope these adjustments provide clarity to the Reviewers and readers of this work.

References

1. Cabral A, Loya B, Alarcon-Segovia D. Bone remodeling and osteophyte formation after remission of rheumatoid arthritis. J Rheumatol. 1989;16(11):1421-7.

2. Kenney H, Peng Y, Bell R, Wood R, Xing L, Ritchlin C, et al. Persistent popliteal lymphatic muscle cell coverage defects despite amelioration of arthritis and recovery of popliteal lymphatic vessel function in TNF-Tg mice following anti-TNF therapy. Scientific Reports. 2022;12(12751).

---

## [Editor Report · Decision Letter 1]

4 Jun 2024

High-Throughput Micro-CT Analysis Identifies Sex-Dependent Biomarkers of Erosive Arthritis in TNF-Tg Mice and Differential Response to Anti-TNF Therapy

PONE-D-24-06132R1

Dear Dr. Awad,

We’re pleased to inform you that your manuscript has been judged scientifically suitable for publication and will be formally accepted for publication once it meets all outstanding technical requirements.

Kind regards,

Sadiq Umar

Academic Editor

PLOS ONE

---

## [Editor Report · Acceptance letter]

26 Jun 2024

PONE-D-24-06132R1 

PLOS ONE

Dear Dr. Awad, 

I'm pleased to inform you that your manuscript has been deemed suitable for publication in PLOS ONE. Congratulations! Your manuscript is now being handed over to our production team.

Kind regards, 

on behalf of

Dr. Sadiq Umar 

Academic Editor

PLOS ONE